# Influence of sex, age, and education on mood profile clusters

**Peter C. Terry**[1]☯*, **Renée L. Parsons-Smith**[2,3]☯, **Rachel King**[4], **Victoria R. Terry**[1,5]

**1** Centre for Health Research, University of Southern Queensland, Toowoomba, Queensland, Australia, **2** School of Psychology and Counselling, University of Southern Queensland, Toowoomba, Queensland, Australia, **3** School of Health and Behavioural Sciences, University of the Sunshine Coast, Sippy Downs, Queensland, Australia, **4** School of Sciences, University of Southern Queensland, Toowoomba, Queensland, Australia, **5** School of Nursing and Midwifery, University of Southern Queensland, Toowoomba, Queensland, Australia

☯ These authors contributed equally to this work.
* peter.terry@usq.edu.au

**Data Availability Statement:** All data files are available from the University of Southern Queensland ePrints database (DOI: 10.26192/57rm-4t21).

## Abstract

In the area of mood profiling, six distinct profiles are reported in the literature, termed the iceberg, inverse iceberg, inverse Everest, shark fin, surface, and submerged profiles. We investigated if the prevalence of the six mood profiles varied by sex, age, and education among a large heterogeneous sample. The Brunel Mood Scale (BRUMS) was completed via the *In The Mood* website by 15,692 participants. A seeded k-means cluster analysis was used to confirm the six profiles, and discriminant function analysis was used to validate cluster classifications. Significant variations in the prevalence of mood profiles by sex, age, and education status were confirmed. For example, females more frequently reported negative mood profiles than males, and older and more highly educated participants had a higher prevalence of the iceberg profile than their younger and lesser educated counterparts. Findings suggest that refinement of the existing tables of normative data for the BRUMS should be considered.

## Introduction

Moods and emotions are pervasive to human functioning and deeply influence an individual's effort, attention, decision-making, memory, behavioural responses, and interpersonal interactions [1, 2]. Mood and emotions are closely related but distinct constructs that, collectively, form the first part of the classic ABC triad (affect, behaviour, cognition) devised by social psychologists to explain human functioning [3]. Although the mood construct has eluded a universally accepted definition among researchers, it is often conceptualised as representing a set of transient feelings that provide the emotional backdrop for interactions with the world around us. Moods are typically of longer duration and lower intensity than emotions, and not always attributable to an identifiable cause [4].

A process referred to as mood profiling, in which an individual's mood scores are plotted against normative scores to create a graphical profile, is often used to identify commonly occurring patterns of mood responses. The Brunel Mood Scale (BRUMS) [5, 6], a derivative of

**Funding:** The author(s) received no specific funding for this work.

**Competing interests:** The authors have declared that no competing interests exist.

the Profile of Mood States (POMS) [7, 8], has become a frequently used psychometric test for quantifying moods. Both the POMS and the BRUMS have been used extensively in the domain of sport and exercise psychology to investigate the antecedents, correlates, and behavioural consequences of moods; in particular, the effects of moods on the performance and psychological well-being of athletes and exercisers [9, 10].

In that context, three distinct mood profiles have been identified, referred to as the iceberg, inverse iceberg, and Everest profiles. The *iceberg profile* is characterized by a high level of vigour, combined with low levels of tension, depression, anger, fatigue, and confusion, and has long been associated with positive mental health and good athletic performance [11, 12]. The *inverse iceberg profile* is characterized by below average vigour, combined with above average tension, depression, anger, fatigue, and confusion, and is associated with underperformance and risk of pathogenesis [13]. A third mood profile, termed the *Everest profile*, is characterized by higher vigour scores than the iceberg profile and lower scores for tension, depression, anger, fatigue, and confusion, and is associated with superior performance [14].

More recently, four novel mood profiles have been identified among the general population, referred to as the inverse Everest, shark fin, surface, and submerged profiles [15]. The *inverse Everest profile* is characterized by low vigour, high tension and fatigue, and very high depression, anger, and confusion; the *shark fin profile* by below average tension, depression, anger, vigour, and confusion, combined with high fatigue; the *surface profile* by average scores on all mood dimensions; and the *submerged profile* by below average scores on all mood dimensions. These four mood profiles, together with the iceberg and inverse iceberg profiles, have been replicated in a variety of contexts, including among heterogeneous samples of English-speaking and Italian-speaking sport participants [16, 17] and among a representative sample of the Singaporean population [18], suggesting that the profile clusters are robust across different language and cultural contexts.

The BRUMS is used in many applied research settings around the globe to, for example, evaluate population-level mental health and monitor the psychological well-being of cardiac rehabilitation patients in Brazil [19, 20]; manage performance anxiety and prevent injuries among adolescent ballet dancers in Japan [21]; screen for risk of post-traumatic stress disorder among military personnel in South Africa [22]; assess adolescents for elevated suicide risk in the USA [23]; and quantify affective responses to music in Australia [24]. Use of the BRUMS is prevalent in sport for a variety of purposes, including prediction of performance; monitoring athlete mindset; assessing mood responses to poor performance, training load, injury, or long haul travel; screening for risk of overtraining, eating disorders and other pathogenic conditions; and as a general catalyst for discussion between athlete and sport psychologist [10].

The purpose of the current study was to investigate the prevalence of mood profile clusters among a large sample of participants, according to their sex, age, and educational status. The findings will inform applied uses of the BRUMS and future investigations of mood responses in a wide range of different contexts. Given the large sample utilized, the findings will also signal whether refinement of existing tables of normative data [25] should be considered.

## Methods

### Participants

A total of 15,692 participants were involved in the study. The sample was socio-demographically heterogeneous, with an approximately equal representation of males (53.8%) and females (46.2%), and a range of age groupings and educational levels (see Table 1). The ethnic composition of the sample was 53.9% Caucasian, 19.0% Asian, 6.0% African, 2.4% Middle Eastern, ~1% Indigenous, with 17.6% selecting the Other ethnicity category. For occupation, the

**Table 1. Participant characteristics and comparison of BRUMS scores by sex, age, and education.**

| Source | n (%) | M | SD | 95% CI |
|---|---|---|---|---|
| Sex [Wilks'$\Lambda$ = .968, $F(6, 15685)$ = 85.70[†], partial $\eta^2$ = .032] | | | | |
| Male | 8,450 (53.8) | | | |
| Tension | | 47.11[†] | 8.60 | [46.92, 47.29] |
| Depression | | 52.09 | 12.07 | [51.83, 52.34] |
| Anger | | 52.44[§] | 10.58 | [52.21, 52.66] |
| Vigour | | 50.41[†] | 9.54 | [50.21, 50.61] |
| Fatigue | | 51.98[†] | 9.39 | [51.78, 52.18] |
| Confusion | | 51.34[†] | 10.54 | [51.11, 51.56] |
| Female | 7,242 (46.2) | | | |
| Tension | | 48.35 | 9.14 | [48.14, 48.56] |
| Depression | | 52.47 | 12.43 | [52.19, 52.76] |
| Anger | | 51.94 | 10.12 | [51.71, 52.17] |
| Vigour | | 48.00 | 9.40 | [47.79, 48.22] |
| Fatigue | | 53.64 | 9.78 | [53.42, 53.87] |
| Confusion | | 52.00 | 11.12 | [51.74, 52.25] |
| Age Group (yr.) [Wilks'$\Lambda$ = .967, $F(24, 54709)$ = 22.23[†], partial $\eta^2$ = .008] | | | | |
| 18–24[a] | 9,765 (62.2) | | | |
| Tension | | 47.66[bde§] | 8.78 | [47.48, 47.83] |
| Depression | | 51.61[bc†] | 11.68 | [51.38, 51.84] |
| Anger | | 51.98[b†] | 10.11 | [51.78, 52.18] |
| Vigour | | 49.62[bc†] | 9.44 | [49.43, 49.81] |
| Fatigue | | 53.18[bde†] | 9.49 | [53.00, 53.37] |
| Confusion | | 51.41[b†] | 10.35 | [51.20, 51.61] |
| 25–35[b] | 3,212 (20.5) | | | |
| Tension | | 48.28[de†] | 9.09 | [47.97, 48.60] |
| Depression | | 53.30 | 12.61 | [52.86, 53.74] |
| Anger | | 52.81 | 10.88 | [52.43, 53.19] |
| Vigour | | 48.16[de†] | 9.82 | [47.82, 48.50] |
| Fatigue | | 52.35[de†] | 9.78 | [52.01, 52.68] |
| Confusion | | 52.69[de†] | 11.76 | [52.29, 53.10] |
| 36–45[c] | 1,348 (8.6) | | | |
| Tension | | 47.67 | 9.17 | [47.18, 48.16] |
| Depression | | 53.95 | 14.15 | [53.20, 54.71] |
| Anger | | 52.74 | 10.93 | [52.16, 53.32] |
| Vigour | | 48.59[de†] | 10.03 | [48.06, 49.13] |
| Fatigue | | 52.77[de†] | 10.15 | [52.22, 53.31] |
| Confusion | | 52.10[e§] | 11.56 | [51.48, 52.72] |
| 46–55[d] | 916 (5.8) | | | |
| Tension | | 46.59 | 8.70 | [46.03, 47.16] |
| Depression | | 53.03[a§] | 13.01 | [52.18, 53.87] |
| Anger | | 52.00 | 10.62 | [51.31, 52.69] |
| Vigour | | 50.13 | 9.35 | [49.52, 50.74] |
| Fatigue | | 50.60 | 9.06 | [50.01, 51.19] |
| Confusion | | 50.62 | 10.98 | [49.91, 51.33] |
| 56+[e] | 451 (2.9) | | | |
| Tension | | 46.23 | 8.42 | [45.45, 47.01] |
| Depression | | 52.57 | 12.53 | [51.41, 53.73] |

*(Continued)*

**Table 1.** (Continued)

| Source | n (%) | *M* | *SD* | 95% CI |
|---|---|---|---|---|
| Anger | | 51.63 | 9.91 | [50.71, 52.54] |
| Vigour | | 50.90 | 8.13 | [50.15, 51.66] |
| Fatigue | | 50.48 | 9.34 | [49.61, 51.34] |
| Confusion | | 49.97 | 10.24 | [49.02, 50.92] |
| **Education** [Wilks' $\Lambda$ = .980, $F$(18, 44359) = 17.68[†], partial $\eta^2$ = .007] | | | | |
| < High School[a] | 883 (5.6) | | | |
| Tension | | 50.02 [bcd†] | 9.92 | [49.36, 50.67] |
| Depression | | 56.14 [bcd†] | 14.92 | [55.16, 57.13] |
| Anger | | 55.64 [bcd†] | 12.28 | [54.83, 56.45] |
| Vigour | | 51.37 [bcd†] | 9.90 | [50.72, 52.03] |
| Fatigue | | 52.95 | 9.76 | [52.31, 53.59] |
| Confusion | | 55.31 [bcd†] | 12.59 | [54.48, 56.14] |
| High School[b] | 6,629 (42.2) | | | |
| Tension | | 47.21[d†] | 8.70 | [47.00, 47.42] |
| Depression | | 51.43[d†] | 11.62 | [51.15, 51.71] |
| Anger | | 51.64[d†] | 9.82 | [51.40, 51.88] |
| Vigour | | 49.08 | 9.48 | [48.85, 49.30] |
| Fatigue | | 52.75 | 9.55 | [52.52, 52.98] |
| Confusion | | 50.90[cd†] | 10.13 | [50.65, 51.14] |
| University[c] | 5,517 (35.2) | | | |
| Tension | | 47.70 | 8.67 | [47.48, 47.93] |
| Depression | | 51.97[d†] | 11.67 | [51.66, 52.28] |
| Anger | | 52.11 | 10.26 | [51.84, 52.38] |
| Vigour | | 49.42 | 9.53 | [49.17, 49.67] |
| Fatigue | | 52.78 | 9.54 | [52.53, 53.04] |
| Confusion | | 51.72 | 10.70 | [51.44, 52.00] |
| Postgraduate[d] | 2,663 (17.0) | | | |
| Tension | | 48.04 | 9.22 | [47.69, 48.39] |
| Depression | | 53.68 | 13.46 | [53.17, 54.20] |
| Anger | | 52.67 | 11.00 | [52.25, 53.09] |
| Vigour | | 48.92 | 9.59 | [48.55, 49.28] |
| Fatigue | | 52.61 | 9.86 | [52.23, 52.98] |
| Confusion | | 52.11 | 11.72 | [51.67, 52.56] |

Superscript letters indicate significant between-group differences

[§]$p < .008$

[†]$p < .001$.

highest proportion of participants indicated Student/Education (32.7%), followed by Athlete/Sport (19.9%). A combined total of 22.9% of respondents ($\leq$ 5% for each category) selected Clerical/Administration, Community/Personal Services, Defence, Health/Medical, Manager/Professional, Manual/Labourer, Operator/Driver, Police/Emergency, Sales/Marketing, or Technical/Trade. Further, 22.5% of the sample selected the Other occupation option and 2.0% of the sample selected Not Currently Employed. In terms of reasons for completing the BRUMS, the highest proportion of the sample selected General Interest (34.2%), followed by Wanting to Help with Research (27.3%), Preparing for a Performance or Task (19.5%), and Not Feeling my Normal Self (3.4%). A total of 15.6% of the sample selected the Other reason option.

## Measures

**Brunel Mood Scale (BRUMS).** Mood responses were assessed using the 24-item BRUMS [5, 6], comprising six subscales (i.e., tension, depression, anger, vigour, fatigue, and confusion) of four items each. Item responses were rated on a 5-point Likert scale of 0 = *not at all*, 1 = *a little*, 2 = *moderately*, 3 = *quite a bit*, and 4 = *extremely*, with total possible subscale scores ranging from 0–16. The response timeframe was "How do you feel right now?" Once the 24 items are condensed into six subscale scores they are treated as scale variables. The psychometric robustness of the BRUMS has been established using multi-sample confirmatory factor analysis across large samples of adult students, adult athletes, young athletes, and schoolchildren [5, 6]. The BRUMS has demonstrated satisfactory internal consistency, with Cronbach alpha coefficients ranging from 0.74–0.90 for the six subscales.

The BRUMS has been translated and validated in several languages and cultural contexts, including Afrikaans [26], Brazilian Portuguese [27], Chinese [28], Czech [29], French [30], Hungarian [31], Italian [31, 32], Japanese [33], Malay [34, 35], Persian [36], Singaporean [37], Serbian [38], and Spanish [39]. Measures derived from the POMS have been criticised for providing a limited assessment of the global domain of mood [40], so researchers using the BRUMS are cautioned not to extrapolate findings beyond the six specific mood dimensions assessed.

***In The Mood* website.** The website [41] facilitates a prompt calculation and interpretation of individual mood responses to the BRUMS and provides respondents with evidence-based mood regulation strategies for each mood dimension, where appropriate. An automated report is generated, including raw and standard scores, reference to normative scores, a graphical representation of the individual mood profile, and suggested mood regulation strategies, where appropriate.

## Procedure

Adult participants ($\geq$ 18 years) were recruited from the general population using a snowballing technique over a 7-year period from March 2011 to January 2018. Informed consent was obtained by clicking on an "I agree" checkbox. This study was carried out in accordance with the recommendations of the Australian Code for the Responsible Conduct of Research. The protocol was approved by the Human Research Ethics Committee at the University of Southern Queensland (approval number: H11REA023, H13REA169, H16REA015). All participants gave written informed consent in accordance with the Declaration of Helsinki. Analyses were conducted using IBM SPSS Statistics for Windows, Version 25 [42].

# Results

## Data screening

All cases were screened for implausible responses (e.g., scoring 0 or 16 on each subscale) and deleted where identified. The *In The Mood* website requires responses to all items prior to submission, and hence no missing values were detected. As found in previous studies of mood [15, 17], significant deviation from univariate normality was evident and expected for some subscales (e.g., depression), being typical for distributions of negative mood scores [5]. It was judged that levels of skewness and kurtosis were unlikely to affect findings of multivariate analysis methods, particularly given the very large sample size. The full range of scores from 0–16 was recorded within the study sample for each of the BRUMS subscales.

## Comparison with normative scores

Descriptive statistics for each of the BRUMS subscales are shown in Table 2, together with statistical comparison of observed mean scores with normative means. Mean values for each

**Table 2. Comparison of mean BRUMS scores *vs.* norms (n = 15,692).**

| Subscale | *M* | *SD* | 95% CI | *t* | *d* |
|---|---|---|---|---|---|
| Tension | 47.68 | 8.88 | [47.54, 47.82] | 12.01[†] | 0.26 |
| Depression | 52.27 | 12.24 | [52.07, 52.46] | 8.89[†] | 0.19 |
| Anger | 52.21 | 10.37 | [52.04, 52.37] | 10.03[†] | 0.21 |
| Vigour | 49.30 | 9.55 | [49.15, 49.45] | 3.41[†] | 0.07 |
| Fatigue | 52.75 | 9.61 | [52.60, 52.90] | 13.32[†] | 0.29 |
| Confusion | 51.64 | 10.81 | [51.47, 51.81] | 7.18[†] | 0.15 |

All scores are T-scores; *t* = t-test for difference between observed mean and normative mean of 50 (*SD* = 10); *d* = effect size

[†]$p < .001$.

mood dimension showed significant deviations from the normative means, although the magnitude of the differences was small in each case [43].

## Cluster analysis

The raw score cluster metrics from Parsons-Smith et al. [15] (Table 3) were used to perform a seeded k-means cluster analysis with a prescribed 6-cluster solution representing the six mood profiles. All six hypothesised clusters were clearly identified. The prevalence of specific mood profiles within the sample (*N* = 15,692), in descending order, was iceberg (28.5%), submerged (23.9%), surface (15.6%), shark fin (15.5%), inverse iceberg (11.8%), and inverse Everest (4.6%; Fig 1). Descriptive statistics for each of the mood profiles are shown in Table 4.

## Discriminant function analysis

A multiple discriminant function analysis (DFA) was conducted to calculate how well the six subscales discriminated between the different clusters, and how well the DFA classification process categorised cases into those clusters. DFA involves a computational process to identify orthogonal dimensions along which naturally occurring groups vary, as well as a classification procedure to predict group memberships [44]. As outliers tend to have greater influence within clusters when applying classification methods, Mahalanobis distances were calculated within each of the clusters to ensure that one group was not more affected than others. The percentage of participants identified as multivariate outliers in each cluster was 2.1% (94/ 4,479, iceberg profile), 1.3% (9/716, inverse Everest profile), 0.4% (7/1,859, inverse iceberg profile), 1.1% (27/2,431, shark fin profile), 2.3% (85/3,753, submerged profile), and 0.8% (19/ 2,454, surface profile).

**Table 3. Raw score cluster centroids from Parsons-Smith et al. [15].**

| Source | Cluster | | | | | |
|---|---|---|---|---|---|---|
| | **1** | **2** | **3** | **4** | **5** | **6** |
| Tension | 1.15 | 10.42 | 6.34 | 1.75 | 1.29 | 4.58 |
| Depression | 0.25 | 11.19 | 5.11 | 1.26 | 0.59 | 1.69 |
| Anger | 0.41 | 10.23 | 4.52 | 0.95 | 0.48 | 2.26 |
| Vigour | 10.62 | 4.69 | 5.98 | 4.14 | 4.72 | 9.10 |
| Fatigue | 2.39 | 11.83 | 8.59 | 9.97 | 2.91 | 4.76 |
| Confusion | 0.54 | 10.75 | 5.84 | 1.32 | 0.90 | 3.27 |

1 = Iceberg Profile, 2 = Inverse Everest Profile, 3 = Inverse Iceberg Profile, 4 = Shark Fin Profile, 5 = Submerged Profile, 6 = Surface Profile.

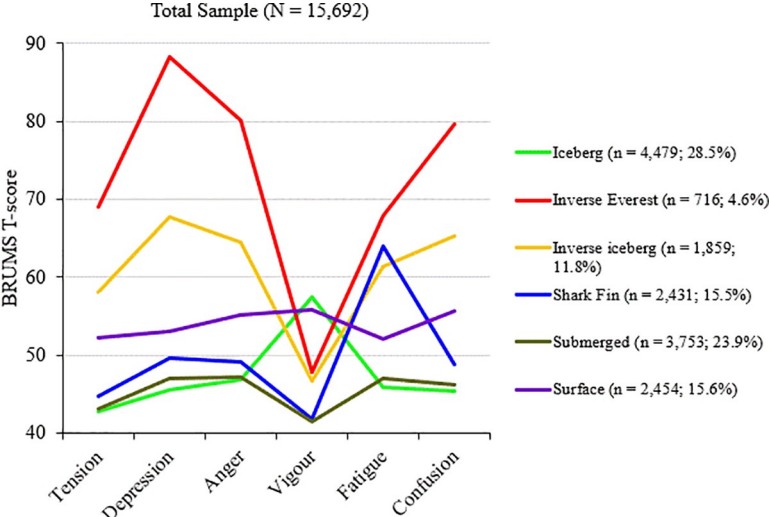

**Fig 1. Graphical representation of the six mood clusters.**

The full data set was split into training and test (validation) sets so that the classification accuracy of the DFA model could be evaluated on a set of data independent of the set used to build the model [45]. Based on the cluster membership of each case identified in the k-means cluster analysis, 20% of cases were randomly selected from each cluster and allocated to the test set ($n = 3,135$). The remaining 80% of cases made up the training set ($n = 12,557$). The ratio of cases to predictor variables (i.e., 2,093 to 1) and the number of cases in the smallest grouping (i.e., 573) far exceeded any published guidelines on case numbers. The canonical correlations for the five discriminant functions were significant ($p < .001$), with the first three discriminant functions accounting for 99.2% of the cumulative total variance (see Table 5). Each of the six original subscales was at least moderately correlated ($> .4$) with at least one of the discriminant functions, indicating that all six mood dimensions contribute to the discrimination between the six clusters [44] (Table 6).

**Table 4. Descriptive statistics of the 6-cluster solution.**

| Source | Iceberg (n = 4,479; 28.5%) | | | Inverse Everest (n = 716; 4.6%) | | | Inverse Iceberg (n = 1,859; 11.8%) | | |
|---|---|---|---|---|---|---|---|---|---|
| | M | SD | 95% CI | M | SD | 95% CI | M | SD | 95% CI |
| Tension | 42.87 | 3.74 | [42.76, 42.98] | 68.97 | 8.11 | [68.37, 69.56] | 58.06 | 7.35 | [57.72, 58.39] |
| Depression | 45.51 | 3.30 | [45.41, 45.61] | 88.23 | 11.00 | [87.42, 89.04] | 67.69 | 9.68 | [67.25, 68.13] |
| Anger | 46.92 | 3.69 | [46.81, 47.03] | 80.19 | 10.21 | [79.45, 80.94] | 64.48 | 9.32 | [64.05, 64.90] |
| Vigour | 57.54 | 5.49 | [57.38, 57.70] | 47.86 | 10.42 | [47.09, 48.62] | 46.72 | 7.76 | [46.36, 47.07] |
| Fatigue | 45.82 | 4.70 | [45.68, 45.95] | 67.92 | 7.49 | [67.38, 68.47] | 61.35 | 7.68 | [61.00, 61.70] |
| Confusion | 45.37 | 3.75 | [45.26, 45.48] | 79.57 | 10.04 | [78.83, 80.31] | 65.35 | 8.33 | [64.97, 65.73] |
| Source | Shark Fin (n = 2,431; 15.5%) | | | Submerged (n = 3,753; 23.9%) | | | Surface (n = 2,454; 15.6%) | | |
| | M | SD | 95% CI | M | SD | 95% CI | M | SD | 95% CI |
| Tension | 44.71 | 5.16 | [44.51, 44.92] | 43.19 | 4.68 | [43.04, 43.34] | 52.22 | 6.48 | [51.97, 52.48] |
| Depression | 49.65 | 6.69 | [49.38, 49.91] | 47.02 | 5.57 | [46.84, 47.20] | 53.04 | 7.02 | [52.76, 53.32] |
| Anger | 49.12 | 5.45 | [48.90, 49.33] | 47.19 | 4.28 | [47.05, 47.32] | 55.14 | 7.70 | [54.83, 55.44] |
| Vigour | 41.90 | 6.95 | [41.62, 42.18] | 41.51 | 5.43 | [41.34, 41.69] | 55.86 | 6.20 | [55.61, 56.11] |
| Fatigue | 63.93 | 6.20 | [63.68, 64.18] | 47.05 | 4.60 | [46.90, 47.19] | 52.09 | 5.81 | [51.86, 52.32] |
| Confusion | 48.83 | 6.37 | [48.58, 49.09] | 46.15 | 4.80 | [46.00, 46.31] | 55.72 | 7.20 | [55.44, 56.01] |

**Table 5. Discriminant functions (n = 12,557).**

| DF | Eigenvalue | Variance Explained (%) | Cumulative Variance (%) | Canonical Correlation | Wilks' Λ | df | $\chi^2$ |
|---|---|---|---|---|---|---|---|
| 1 | 6.563 | 75.7 | 75.7 | .932 | .032 | 30 | 43086.82[†] |
| 2 | 1.514 | 17.5 | 93.2 | .776 | .244 | 20 | 17695.20[†] |
| 3 | .520 | 6.0 | 99.2 | .585 | .614 | 12 | 6127.95[†] |

DF = discriminant function; $R^2$ = percentage of between-group variance; df = degrees of freedom

[†]$p < .001$.

The sample was random, so the group sizes were considered the best estimates of the population proportions, and the prior probabilities were calculated accordingly. The prior probabilities (i.e., the estimated likelihood that a case belongs to a particular group) for the iceberg, inverse Everest, inverse iceberg, shark fin, submerged, and surface profiles were 28.5%, 4.6%, 11.8%, 15.5%, 23.9%, and 15.5%, respectively. The classification procedure performed on the test set found that 94.3% of cases were correctly classified into their original clusters. This figure was notably higher than the minimum classification accuracy rate of 45.3%, which was computed by squaring and summing the proportional by chance accuracy rates + 25%. The percentage of correct classifications were iceberg profile = 99.9%, inverse Everest profile = 90.2%, inverse iceberg profile = 92.7%, shark fin profile = 88.7%, submerged profile = 97.1%, and surface profile = 87.8%. The high hit ratio is consistent with percentages reported by Parsons-Smith et al. [15], being 95.2%, 94.7%, and 95.2% for each of the three samples investigated, which shows evidence of a high degree of consistency in the classification scheme (see Table 7).

## Mood responses by sex, age, and education

Three between-group MANOVAs with *post-hoc* pairwise comparisons were conducted to establish whether mood scores varied by sex, age grouping, and education status. Effect sizes in the form of partial $\eta^2$ were calculated for each significant pairwise comparison to assess explained variance [43]. For each demographic grouping, there was a significant multivariate main effect on a composite of the six dependent variables (Table 1).

Significant univariate main effects were also identified. For sex, an examination of the mean scores for each dependent variable (Table 1) showed that tension, fatigue, and confusion scores were higher among females, whereas anger and vigour scores were higher among males. No sex differences were found for depression.

Several age group differences were found (Table 1). Participants aged 25–35 yr. reported higher tension, depression, anger, and confusion scores, combined with lower vigour and

**Table 6. Structure matrix (n = 12,557).**

| Mood Dimension | Structure Matrix | | | | |
|---|---|---|---|---|---|
| | 1 | 2 | 3 | 4 | 5 |
| Vigour | - | .816* | .565 | - | - |
| Fatigue | .426 | −.534 | .729* | - | - |
| Depression | .624 | - | - | .693* | - |
| Tension | .494 | - | - | −.573* | - |
| Confusion | .568 | - | - | - | −.668* |
| Anger | .521 | - | - | - | .598* |

*Largest absolute correlation between each variable and any discriminant function.

**Table 7. Cluster classifications (n = 3,135).**

| Cluster | Predicted Group Membership | | | | | | n |
|---|---|---|---|---|---|---|---|
| | 1 | 2 | 3 | 4 | 5 | 6 | |
| Iceberg | 894 | 0 | 0 | 0 | 1 | 0 | 895 |
| Inverse Everest | 0 | 129 | 14 | 0 | 0 | 0 | 143 |
| Inverse Iceberg | 0 | 1 | 344 | 0 | 8 | 18 | 371 |
| Shark Fin | 1 | 0 | 3 | 431 | 44 | 7 | 486 |
| Submerged | 1 | 0 | 4 | 0 | 728 | 17 | 750 |
| Surface | 48 | 0 | 7 | 5 | 0 | 430 | 490 |

1 = Iceberg Profile, 2 = Inverse Everest Profile, 3 = Inverse Iceberg Profile, 4 = Shark Fin Profile, 5 = Submerged Profile, 6 = Surface Profile.

fatigue scores than those aged 18–24 yr. Lower tension and fatigue scores were reported by the 46–55 yr. and 56+ yr. groups than those aged 18–24 yr. and 25–35 yr., and lower confusion scores than the 25–35 yr. group. Vigour scores for the 46–55 yr. and 56+ yr. groups were higher than the 25–35 yr. and 36–45 yr. groups, and those aged 18–24 yr. reported higher vigour scores than the 36–45 yr. group. Depression scores for the 46–55 yr. group was also higher than those aged 18–24 yr. Lower scores for fatigue and confusion were reported by the 56+ yr. group than the 36–45 yr. group. Lower scores for fatigue was also reported by the 46–55 yr. group than the 36-45 yr. group.

Group differences were also found for education status (Table 1). Tension, depression, anger, vigour, and confusion scores for the less than high school group were higher than for all other groups. The postgraduate group reported higher depression scores than all other groups, and higher tension, anger, and confusion scores than the high school group. The university group reported higher confusion scores than the high school group. No education differences in fatigue scores were found.

## Prevalence of specific mood profiles by sex, age, and education

A series of chi-square goodness-of-fit tests were performed to determine whether the inverse iceberg, inverse Everest, surface, iceberg, shark fin, and submerged mood profiles varied according to distributions of sex, age, and level of education. The distribution of mood profile clusters varied significantly from expected values for each grouping variable, indicating associations between mood profiles and sex, age, and education. Cluster distributions can be found in Table 8 and are shown graphically in Figs 2–4.

For sex, males were over-represented for the iceberg and surface profiles, whereas females were over-represented for the inverse Everest, inverse iceberg, shark fin, and submerged profiles. For age group, the two oldest groups (46–55 yr., 56+ yr.) were over-represented for the iceberg profile, whereas the two youngest groups (18–24 yr., 25–35 yr.) were under-represented. The 25–35 yr. and 36–45 yr. groups were over-represented for the inverse Everest profile, whereas the 18–24 yr. group was under-represented. The 25–35 yr. group was similarly over-represented for the inverse iceberg profile, whereas the 46–55 yr. group was under-represented. For the shark fin profile, the 18–24 yr. group was over-represented, whereas the 25–35 yr., 46–55 yr., and 56+ yr. groups were under-represented. For the submerged profile, the 18–24 yr. group was under-represented, whereas the 25–35 yr. group was over-represented. Finally, for the surface profile, the 25–35 yr. and 56+ yr. groups were under-represented, whereas the 18–24 yr. group was over-represented.

For education, the postgraduate group was over-represented for the iceberg profile, whereas the less than high school group was under-represented. The less than high school and

**Table 8. Distribution of mood profile clusters by sex, age, and education (n = 15,692).**

| | Cluster | | | | | |
|---|---|---|---|---|---|---|
| Source | 1 | 2 | 3 | 4 | 5 | 6 |
| **Sex** [$\chi^2(5) = 203.42^\dagger$] | | | | | | |
| Male | 2,752$^{\dagger+}$ | 345$^{\S-}$ | 955$^{*-}$ | 1,122$^{\dagger-}$ | 1,885$^{\dagger-}$ | 1,391$^{\S+}$ |
| Female | 1,727$^{\dagger-}$ | 371$^{\S+}$ | 904$^{*+}$ | 1,309$^{\dagger+}$ | 1,868$^{\dagger+}$ | 1,063$^{\S-}$ |
| **Age Group (yr.)** [$\chi^2(20) = 199.71^\dagger$] | | | | | | |
| 18–24 | 2,667$^{\dagger-}$ | 383$^{\dagger-}$ | 1,130 | 1,658$^{\dagger+}$ | 2,269$^{\S-}$ | 1,658$^{\dagger+}$ |
| 25–35 | 869$^{*-}$ | 181$^{\dagger+}$ | 433$^{\S+}$ | 450$^{\S-}$ | 856$^{\dagger+}$ | 423$^{\dagger-}$ |
| 36–45 | 415 | 87$^{\dagger+}$ | 158 | 190 | 303 | 195 |
| 46–55 | 338$^{\dagger+}$ | 46 | 86$^{*-}$ | 101$^{\dagger-}$ | 219 | 126 |
| 56+ | 190$^{\dagger+}$ | 19 | 52 | 32$^{\dagger-}$ | 106 | 52$^{*-}$ |
| **Education** [$\chi^2(15) = 217.47^\dagger$] | | | | | | |
| < High School | 224$^{*-}$ | 95$^{\dagger+}$ | 132$^{\S+}$ | 89$^{\dagger-}$ | 163$^{\dagger-}$ | 180$^{\dagger\dagger+}$ |
| High School | 1,884 | 321$^{\dagger-}$ | 714$^{\dagger-}$ | 1,105$^{\dagger+}$ | 1,689$^{\dagger+}$ | 1,006 |
| University | 1,567 | 222$^{*-}$ | 656 | 880 | 1,270 | 922$^{\S+}$ |
| Postgraduate | 804$^{*+}$ | 168$^{\dagger+}$ | 357$^{\S+}$ | 357$^{\dagger-}$ | 631 | 346$^{\dagger-}$ |

1 = Iceberg Profile (*n* = 4,479), 2 = Inverse Everest Profile (*n* = 716), 3 = Inverse Iceberg Profile (*n* = 1,859), 4 = Shark Fin Profile (*n* = 2,431), 5 = Submerged Profile (*n* = 3,753), 6 = Surface Profile (*n* = 2,454)

$^{+}$over-represented, $^{-}$under-represented

$^{*}$*p* < .05

$^{\S}$*p* < .01

$^{\dagger}$*p* < .001.

postgraduate groups were both over-represented for the inverse Everest profile, whereas the high school and university groups were under-represented. Similarly, the less than high school and postgraduate groups were both over-represented for the inverse iceberg profile, whereas the high school group was under-represented. Conversely, for the shark fin profile, the less

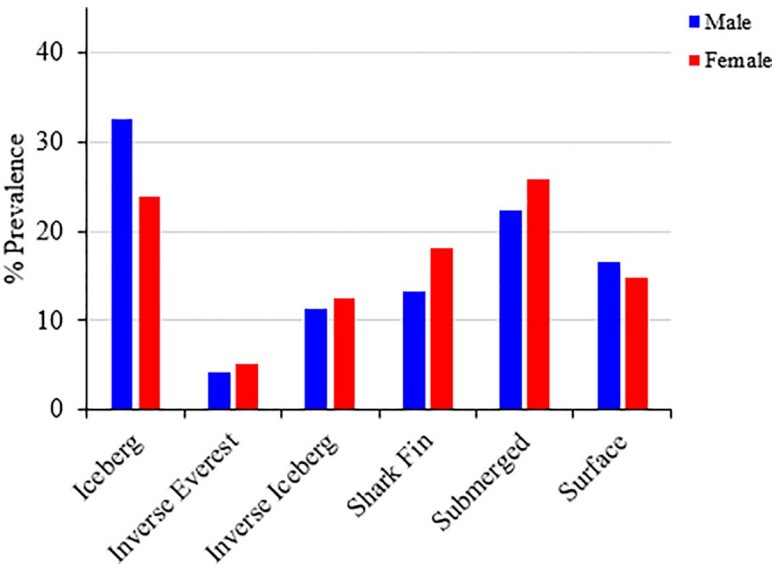

**Fig 2. Prevalence of clusters by sex (n = 15,692).**

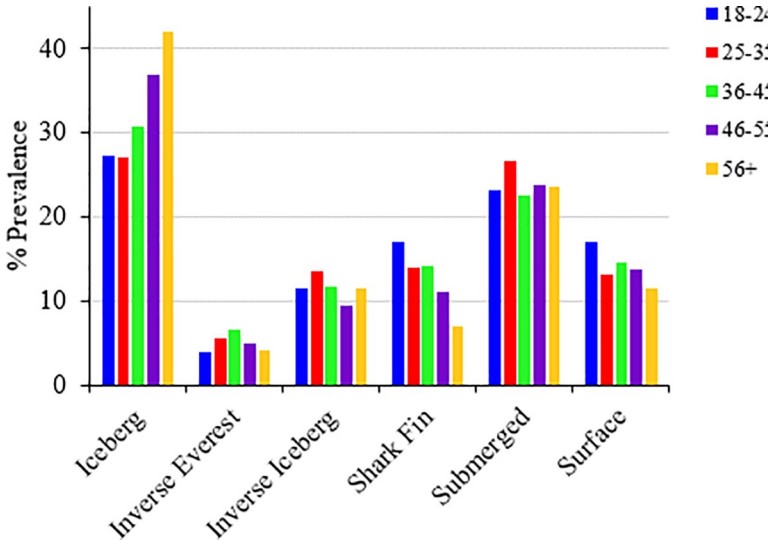

**Fig 3. Prevalence of clusters by age (n = 15,692).**

than high school and postgraduate groups were both under-represented, whereas the high school group was over-represented. For the submerged profile, the high school group was over-represented, whereas the less than high school group was under-represented. Finally, for the surface profile, the less than high school and university groups were over-represented, whereas the postgraduate group was under-represented.

## Discussion

Theoretically, especially given the large sample, mean scores for all six mood dimensions should have approximated a T-score of 50. However, it was shown that all subscale means deviated significantly from normative means, although effect sizes were uniformly small (Table 2). This finding provides a rationale for revisiting existing tables of normative values for the BRUMS [25] with the aim of refining them to better reflect the current dataset, which is the largest and most representative available.

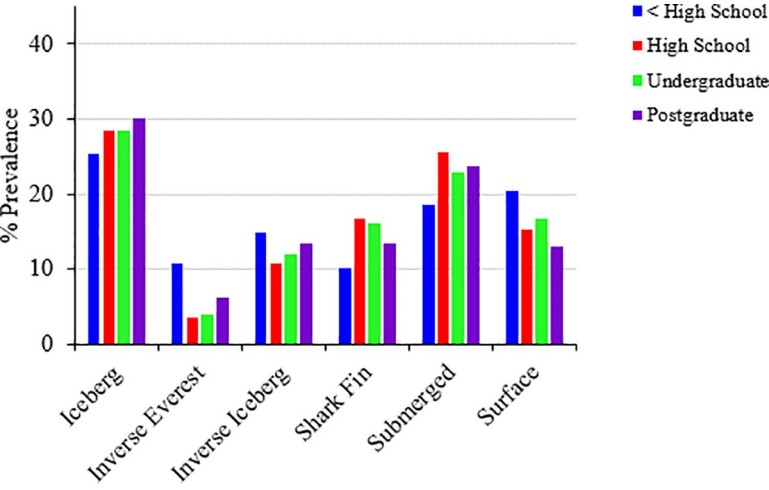

**Fig 4. Prevalence of clusters by education (n = 15,692).**

Between-group comparisons identified similar results to those reported previously. In common with the current findings, previous research on mood profile clusters has consistently identified a higher prevalence of the iceberg profile among males and a higher prevalence of the negative mood profiles among females [15, 17, 18]. The Australian Bureau of Statistics [46] notes that mood disorders are almost twice as prevalent among females than males (8.4% vs. 4.3%) and, globally, women are approximately twice as likely as men to suffer from a mental illness [47]. Assorted biochemical, chronobiological, neurological, psychological, and psychosocial explanations have been advanced to explain sex differences in mood responses. For example, mood disturbance among females has been linked to estrogen-serotonin interactions and cyclic and/or acute hormone fluctuations associated with reproductive-related events (e.g., menarche, menstruation, pregnancy, postpartum, menopause) [48, 49]. Sex differences in brain structure and subsequent responses to stress [50] have also been identified, as have sex differences in ability to downregulate negative feeling states effectively [51]. Finally, social issues involving sex discrimination, wage differentiation, and inequality in the workforce have all been associated with disparity in prevalence rates of mood disorders [52].

Consistent with some previous findings [15, 17, 18], the prevalence of iceberg profile tended to increase with age, whereas the prevalence of shark fin profile tended to decrease with age (Fig 3). Nuanced differences in use of emotion-regulation strategies may partially explain these age-related variations in reported mood. Younger adults are more likely to engage in maladaptive coping strategies, such as rumination, avoidance, and suppression, all of which are associated with poorer mental health outcomes [51]. In a similar vein, mindfulness has been shown to facilitate effective emotion regulation [53] and psychological well-being [54], and previous age-diverse research has identified that older individuals are more likely to be classified into a high mindfulness profile than younger counterparts [55]. Other age-related trends in our findings were difficult to discern except that the prevalence of the surface profile, which approximates to an average or normal profile, tended to decline with age (Fig 3).

Findings related to level of education showed a trend of the iceberg profile increasing in prevalence among the more highly-qualified participants, whereas the prevalence of the more negative inverse Everest and inverse iceberg profiles was highest among participants with the lowest level of educational attainment (Fig 4). It is possible to speculate on why level of education appears to be associated with more positive mood profiles. Decades of research has demonstrated the positive benefits of education, based on the notion that educational attainment develops higher level skills, leading to higher rates of employment, higher productivity, and higher lifetime earnings for individuals [56]. For example, in Australia, those with a postgraduate degree are much more likely to be in the higher echelons of income earners. Moreover, those with higher levels of education are more likely to be employed, and less likely to experience financial stress [57]. Financial inequality has been shown to impact upon health, including mental health [52, 58]. Hence, better employment and financial circumstances are likely to be associated with more positive moods, although the relationship is by no means linear.

We acknowledge some limitations of our study. Firstly, completion of any online test, including the BRUMS, requires access to a computer with Internet access, which inevitably serves to reduce participation by those from lower socio-economic and marginalised groups. Secondly, the demographic characteristics of our sample showed an over-representation of university-educated participants, which may limit the generalisability of the findings. Participants with a university qualification made up 52.2% of our sample, whereas the percentage in OECD countries is typically less than 40% [56].

In conclusion, our data showed that sex, age group, and level of education all moderated responses to the BRUMS among online respondents. Small but significant differences between observed mean scores and normative means on all subscales suggest that existing tables of

normative data should be refined [59]. The observed differences in mood responses by sex, age, and education raise the question of whether separate tables of normative data should be generated for males and females, for example, as a part of the process of norm refinement. More representative normative data will improve the precision of investigations into the antecedents and behavioural consequences of the mood responses of individual and groups.

In summary, the main contributions of our work to the field of study are (1) the identification of significant differences in mood responses by sex, age, and education among a large online sample, (2) the identification of significant differences between overall sample means and existing normative data on six dimensions of mood, and (3) the provision of further evidence of the robustness of the six mood profile clusters identified by Parsons-Smith and colleagues [15].

## Author Contributions

**Conceptualization:** Peter C. Terry, Renée L. Parsons-Smith, Victoria R. Terry.

**Data curation:** Renée L. Parsons-Smith, Rachel King.

**Formal analysis:** Peter C. Terry, Renée L. Parsons-Smith, Rachel King.

**Funding acquisition:** Peter C. Terry, Victoria R. Terry.

**Investigation:** Peter C. Terry, Renée L. Parsons-Smith, Victoria R. Terry.

**Methodology:** Peter C. Terry, Renée L. Parsons-Smith.

**Project administration:** Renée L. Parsons-Smith, Victoria R. Terry.

**Resources:** Renée L. Parsons-Smith, Rachel King.

**Software:** Renée L. Parsons-Smith, Rachel King.

**Supervision:** Peter C. Terry.

**Validation:** Renée L. Parsons-Smith, Rachel King, Victoria R. Terry.

**Visualization:** Peter C. Terry, Renée L. Parsons-Smith.

**Writing – original draft:** Peter C. Terry, Renée L. Parsons-Smith, Rachel King.

**Writing – review & editing:** Peter C. Terry, Renée L. Parsons-Smith, Rachel King, Victoria R. Terry.

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
