## [Decision Letter · Decision Letter 0]

15 Dec 2020

PONE-D-20-35037

Influence of Sex, Age, and Education on Mood Profile Clusters

PLOS ONE

Dear Dr. Terry,

Thank you for submitting your manuscript to PLOS ONE. After careful consideration by the reviewer and myself as a second reviewer, we feel that it has merit but does not fully meet PLOS ONE’s publication criteria as it currently stands. Therefore, we invite you to submit a revised version of the manuscript that addresses the only 2 points raised during the review process: a) see Reviewer's comments; b) Please, highlight in a couple of lines at the end of discussion what is your main contribution to the field through this work.

We look forward to receiving your revised manuscript.

Kind regards,

Juan-Carlos Pérez-González, Ph.D.

Academic Editor

PLOS ONE

Journal Requirements:

Reviewers' comments:

Reviewer's Responses to Questions

**Comments to the Author**

1. Is the manuscript technically sound, and do the data support the conclusions?

Reviewer #1: Yes

2. Has the statistical analysis been performed appropriately and rigorously? 

Reviewer #1: Yes

3. Have the authors made all data underlying the findings in their manuscript fully available?

Reviewer #1: Yes

4. Is the manuscript presented in an intelligible fashion and written in standard English?

Reviewer #1: Yes

5. Review Comments to the Author

Reviewer #1: The authors have made an excellent analysis of the influence of sex, age, and education on mood profile. The results will improve the accuracy of future research in this area of knowledge.

Only one area needs revision:

The authors should explain, in more detail, some characteristics of the participants: ethnicity, occupation, reason to complete the mood assessment.

6. PLOS authors have the option to publish the peer review history of their article (what does this mean?). If published, this will include your full peer review and any attached files.

Reviewer #1: No

---

## [Author Response · Author response to Decision Letter 0]

22 Dec 2020

PONE-D-20-35037

Influence of Sex, Age, and Education on Mood Profile Clusters

Response to Editor

1. We have attempted to ensure that the manuscript now meets PLOS ONE style requirements.

2. The DOI required to access our data will be provided once the paper is accepted for publication.

3. The ORCID for the corresponding author has been validated in Editorial Manager.

4. The ethics statement now appears only in the Methods section of our manuscript.

Response to Reviewer

5. As requested, we have explained the characteristics of the participants in greater detail as they relate to ethnicity, occupation, reason to complete the mood assessment (see lines 96-108).

6. The main contributions of our work have been added at the end of the Discussion (see lines 382-387).

7. Figures have been uploaded to PACE as tif files.

Regards

Dr Peter C Terry

---

## [Editor Report · Decision Letter 1]

29 Dec 2020

Influence of Sex, Age, and Education on Mood Profile Clusters

PONE-D-20-35037R1

Dear Dr. Terry,

We’re pleased to inform you that your manuscript has been judged scientifically suitable for publication and will be formally accepted for publication once it meets all outstanding technical requirements.

Kind regards,

Juan-Carlos Pérez-González, Ph.D.

Academic Editor

PLOS ONE

---

## [Editor Report · Acceptance letter]

22 Jan 2021

PONE-D-20-35037R1 

Influence of sex, age, and education on mood profile clusters 

Dear Dr. Terry:

I'm pleased to inform you that your manuscript has been deemed suitable for publication in PLOS ONE. Congratulations! Your manuscript is now with our production department. 

Kind regards, 

on behalf of

Dr. Juan-Carlos Pérez-González 

Academic Editor

PLOS ONE